# Cavitation-Mediated Immunomodulation and Its Use with Checkpoint Inhibitors

**DOI:** 10.3390/pharmaceutics15082110

**Published:** 2023-08-09

**Authors:** Matilde Maardalen, Robert Carlisle, Constantin Coussios

**Affiliations:** Institute of Biomedical Engineering, Department of Engineering Science, University of Oxford, Oxford OX1 3PJ, UK

**Keywords:** ultrasound, cavitation, cancer, immunotherapy, checkpoint inhibitor

## Abstract

The promotion of anti-tumour immune responses can be an effective route to the complete remission of primary and metastatic tumours in a small proportion of patients. Hence, researchers are currently investigating various methods to further characterise and enhance such responses to achieve a beneficial impact across a wider range of patients. Due to its non-invasive, non-ionising, and targetable nature, the application of ultrasound-mediated cavitation has proven to be a popular method to enhance the delivery and activity of immune checkpoint inhibitors. However, to optimise this approach, it is important to understand the biological and physical mechanisms by which cavitation may promote anti-tumour immune responses. Here, the published literature relating to the role that cavitation may play in modulating anti-tumour immunity is therefore assessed.

## 1. Cancer Immunology

The interplay between tumours and the immune system was suggested in the mid-19th century when Rudolf Virchow hypothesised that cancers arise at sites of chronic inflammation [1]. A few decades later, Paul Ehrlich proposed that the immune system plays a critical role in protecting the host from cancer development [2]. Indeed, it is now clear that the immune system plays a dual role in cancer progression [3]. Thus, to best harness the immune system’s anti-tumour role it is essential to understand how a tumour-specific immune response may arise, how the tumour evades it, and how therapeutic intervention can most effectively overcome such evasion.

### 1.1. Cancer-Immunity Cycle

Chen and Mellman described the cancer-immunity cycle—events necessary for an anti-tumour immune response to lead to effective killing of cancer cells [4]. Firstly, Tumour-associated antigens (TAAs) or Tumour-specific antigens (TSAs) need to be released and captured by tissue-resident Antigen-presenting cells (APCs) such as Dendritic cells (DCs). The phagocytosed antigens are processed into small peptides and presented on Major histocompatibility complex (MHC) I or II [5]. Additionally, immunogenic signals such as proinflammatory cytokines released by dying tumour cells are essential to induce DC maturation and turn them into potent APCs. Activated DCs increase their synthesis of MHC molecules and begin to express the co-stimulatory molecules CD80 and CD86 on their surface [4].

Next, DCs migrate to lymph nodes, along with their cargo, to activate an adaptive immune response. DCs present the captured antigen on MHC I or II to naive CD8+ or CD4+ T cells, respectively [6]. Exogenous antigens are presented on MHC II molecules but also on MHC I molecules via a process called cross-presentation. The cross-presentation of exogenous antigens is critical for anti-tumour immunity [7]. Binding to the peptide/MHC complex and co-stimulatory molecules on the DC surface stimulates the priming and activation of effector and memory T cell responses against the antigen. The activation of CD8+ T cells with specificity for TAAs or TSAs leads to cytotoxic reactions that cause tumour cell death, while activated CD4+ T cells produce cytokines. Activated effector T cells traffic to and infiltrate the tumour. Here, they bind to cancer cells via the T cell receptor presented on T cells and the antigen bound to the MHC molecule on cancer cells, and eventually kill the cancer cells. This releases additional TAAs or TSAs, and the cycle is repeated and amplified, leading to effective clearance of the primary site, potential abscopal effects against other deposits, and a durable defence against recurrence [4].

Each step in the cancer-immunity cycle is an important contributor to effective killing of cancer cells. Hence, when one or more of these steps are blocked it may lead to an insufficient anti-tumour immune response.

### 1.2. Evading Immune Recognition

In cancer patients, the cancer-immunity cycle does not perform optimally. Indeed, the progression of tumour growth is dependent on an escape from anti-tumour immune responses. Such an escape is achieved by hiding from immune cells, excluding them, or down-regulating their activity.

Tumour cells can hide from immune cells by altering their surface molecule expression. This may be achieved by down-regulating the expression of MHC molecules on tumour cells, particularly MHC I. This limits the ability of Cytotoxic T lymphocytes (CTLs) to recognise tumour cells and initiate apoptosis [8]. However, the absence of MHC molecules often leads to Natural killer (NK) cell activation, but these innate immune cells have no immunological memory and will therefore not provide a sustained anti-tumour immune response [9]. Furthermore, the tumours that have normal expression of MHC molecules often lack co-stimulatory molecules such as CD80 or CD86, which abrogates the activation of T lymphocytes. Lastly, cancer cells are often unrecognised by CTLs because tumour variants with low TAA or TSA production have been selected [10].

The irregular tumour vasculature, elevated interstitial fluid pressure, and dense extracellular matrix combine to restrict the movement of immune cells into and throughout the tumour [11]. Indeed, the abnormal blood and lymphatic vasculature in tumours function as both physical and functional barriers: compressed blood vessels limit the tumour infiltration of cytotoxic immune cells, whereas dysfunctional lymphatic vessels hinder APCs from migrating to lymph nodes and activating T cells [12].

Tumour cells also deploy multiple mechanisms to remodel the Tumour microenvironment (TME) into an immunosuppressive phenotype. Firstly, the TME contains different immune suppressive cells such as Myeloid-derived suppressor cells (MDSCs) and Regulatory T (Treg) cells. MDSCs migrate from the tumour to the lymph nodes, spleen, and peripheral vessels, where they inhibit activation of tumour-specific T lymphocytes [13]. Treg lymphocytes suppress CD4+ and CD8+ T cells by producing specific cytokines such as Interleukin (IL)-10, IL-4, and IL-13. Moreover, Treg cells can inhibit subpopulations of B lymphocytes and NK cells [10]. Another mechanism involves cancer cells modulating immune checkpoint pathways to evade attack by immune cells [14,15]. Immune checkpoint pathways are important for maintaining self-tolerance by modulating T cell function and protecting surrounding tissue by suppressing immune responses, thereby preventing auto-immunity in healthy individuals. Many immune checkpoints are initiated by ligand–receptor interactions; thus, tumour cells can alter this interaction to their benefit. There is an array of different types of immune checkpoint pathways and mediators, in which the most well characterised are Programmed cell death protein 1 (PD-1), Programmed death ligand 1 (PD-L1), and Cytotoxic T lymphocyte-associated protein 4 (CTLA-4). PD-1 is a cell surface receptor presented on T cells that binds to PD-L1 and PD-L2. The interaction of PD-1 with PD-L1 blocks T cell responses to prevent excessive T cell activation [14]. Hence, tumours expressing PD-L1 or PD-L2 ligands can suppress the T cell-mediated anti-tumour immune response. CTLA-4 is important in the early stages of the immune response. This molecule is expressed on tumour cells and competes with CD28 in binding to CD80 and CD86 ligands on APCs. Consequently, further T cell activation is inhibited [15].

In summary, tumours have multiple methods to evade immune recognition. Consequently, there are several ways in which an anti-tumour immune response may be reactivated. Indeed, immunotherapy is a modality that includes a range of treatment methods, including (1) cancer vaccines, which aim to activate CTLs [16]; (2) CAR T cells, which are patient-derived T cells with a chimeric antigen receptor that are engineered to recognise and bind specific antigens on the surface of cancer cells [17]; and (3) Immune checkpoint inhibitors (ICIs), which interact with immune checkpoints, thereby preventing immune tolerance [18]. The effectiveness of these treatments has been proven in a range of cancers [19,20,21]. In fact, the use of ICIs demonstrated that 12 out of 12 patients with rectal cancer had a clinical complete response [22]. However, it is notable that the greatest ICI successes have come in patient populations that have been carefully selected for an immune “hot” rather than “cold” phenotype [22]. For more widespread impact, methods to optimise such approaches are needed, and driving tumours towards a “hot” phenotype is thought to be an effective way of achieving this. Recent research suggests that Ultrasound (US)-mediated cavitation can result in the immunomodulation of the TME, thus partly reversing tumour immune evasion. Hence, this may be a promising strategy to promote tumour-specific immune responses and enhance the efficacy of ICIs.

## 2. Cavitation-Mediated Immunomodulation

Several studies have served to demonstrate that radiation or mechanically induced tumour cell damage can be used in combination with cancer immunotherapies to provide improved outcomes [23,24,25]. Indeed, the approach of combining non-ionising sources, such as US, to thermally ablate tumour tissue has garnered recent attention [26,27,28]. However, tumour antigens released from thermally ablated tumours are often denatured and may not be effectively presented to the immune system [29]. Additionally, thermal diffusion can damage surrounding healthy tissue, causing adverse effects [30].

US can also be used to induce non-ablative mechanical effects in tumours. Acoustic cavitation is used to describe the expansion and collapse of gas bubbles in response to the alternating compressional and rarefactional US wave. Notably, providing agents to nucleate such acoustic cavitation, i.e., by adding pre-existing gas bubbles—referred to henceforth as Cavitation agents (CAs), can lower the pressure amplitude required to achieve cavitation and consequently reduce the energy deposition within tissue. The volumetric changes can lead to various mechanical phenomena that can impact the surroundings. First, the gentle oscillation of CAs will cause momentum transfer from the bubble to the surrounding liquid, called microstreaming [31]. This can result in shear stresses impacting on cell membranes [32]. More violent oscillation and the collapse of CAs near a surface can produce shock waves and microjets, leading to cellular injury and tissue destruction [31,32]. The scale of the membrane disruption and damage, and the underlying mutations to apoptotic pathways within the cells, will determine the consequent route to death (apoptosis or necrosis) and the resulting immune response. Hence, due to its potential, studies have investigated the immunological effects of acoustic cavitation, which are summarised in Figure 1. Figure 1 highlights that cavitation events may modulate tumour immunity in three ways by the destructive release of TSAs, TAAs, and Damage-associated molecular patterns (DAMPs), by vascular modification, and lastly by altering the cellular activity of cells important for immune responses.

This review therefore focuses on how acoustic cavitation, used without the intention of causing heating, may promote anti-tumour immune responses. The evidence regarding the role that cavitation may play in the immune outputs, identified in Figure 1, is assessed. Details of the treatment protocols and US parameters of the papers mentioned in this section can be found in Table 1.

### 2.1. Release of Tumour-Associated Antigens and Damage-Associated Molecular Patterns

The release of TAAs, TSAs, and DAMPs is an important part of activating an anti-tumour immune response. DAMP molecules, including High-mobility-group box 1 (HMGB1), Heat shock proteins (HSPs), calreticulin, and Adenosine triphosphate (ATP), are danger signals that are often released when cells are exposed to stress. These danger signals can promote an immune response by initiating various mechanisms such as stimulating neutrophils or monocytes to produce and secrete proinflammatory cytokines and chemokines, and up-regulate co-stimulatory molecules [38]. In regards to anti-tumour immune responses, cancer cells undergoing immunogenic cell death may expose DAMPs on their surface, which helps DCs engulf the cancer cells. This then leads to the presentation of tumour antigens followed by CTL responses [39]. Thus, the release of DAMPs ultimately leads to the activation of anti-tumour immunity. Researchers have therefore sought to characterise how cavitation may cause the release of DAMPs and tumour antigens, and subsequent immune responses in tumours.

Liu et al. investigated immunological changes induced by US+CA in a colon tumour (CT26) murine model [40]. Immune cell populations were studied at days 1, 3, and 18 after treatment and compared to untreated tumours. The analyses revealed a significant (*p* < 0.05) increase in percentage of CD8+ T cells among CD45+ leukocyte cells across all three days; however, the percentage of CD4+Foxp3+ cells (Tregs) relative to CD45+ leukocytes remained stable. The authors confirmed, by immunohistochemical staining, that US+CA triggered the production of HSP60, although the level was not quantified. A link was therefore suggested between the release of this DAMP and the increased CTL infiltration observed. Additionally, these authors measured temperature in a tissue-mimicking phantom using similar conditions as in vivo, and no increase was observed upon US+CA treatment, indicating that the release of HSP60 was most likely due to mechanical stress and not heating [40].

In a similar study carried out by Joiner et al., pancreatic tumours (KPC) were either untreated or treated with US+CA and excised 2 or 15 days after treatment [41]. A non-significant (*p* = 0.053) 1.6-fold increase of the HMGB1 level, as measured by immunohistochemistry, was detected 2 days after US+CA treatment compared to untreated tumours. However, no significant difference in cell death marker cleaved caspase-3 or HSP70 was observed between the two groups. Furthermore, tumour draining lymph nodes exhibited a significant (*p* < 0.05, *p* < 0.005, and *p* < 0.005, respectively) increase in the number of F4/80+CD11b+ cells (macrophages), CD8+ cells (CTLs), and CD4+ cells (T helper cells) 2 days after treatment. It was concluded that this was a transient immune response due to the majority of these effects subsiding 15 days post-treatment. Notably, US-only or CA-only controls were not included in this study [41].

Furthermore, Hu et al. assessed the release of both TAAs and DAMPs after US+CA, US-only, or CA-only treatment, as well as no treatment [42]. RM1-OVA cells were used to permit measurement of the release of Ovalbumin (OVA) and DAMPs in vitro. At both 0 and 4 h after US+CA treatment, HMGB1, calreticulin, and HSPA2 levels were increased compared to the other treatment groups, though the scale of this increase was not quantified. ATP release was significantly (*p* = 0.004) higher for the US+CA-treated cells. This DAMP release was attributed to the increase in tumour cell necrosis observed both in vitro and in vivo in the US+CA group. The RNAseq data of the RM1 tumour models confirmed that the TME exhibited a high level of pathways related to ’acute inflammation response’ and ’cell death’, suggesting that US+CA can cause tumour cell injury. Hu et al. also investigated and compared the immune microenvironment in vivo after US+CA versus no treatment. They showed a significant (*p* < 0.05, *p* < 0.005, *p* < 0.005, *p* < 0.005, and *p* < 0.005, respectively) increase in CD45+ cells (leukocytes), CD11c+ cells (DCs), CD11b+ (monocytes), 49b+ cells (NK cells), and CD68+ cells (macrophages) both 1 day and 4 days after treatment with US+CA. Further, it was concluded that CD8+ T cells were the main contributors to the US+CA treatment effects. This was due to the fact that the tumour volume after treatment with US+CA with the addition of CD4+ and CD8+ T cell-depleting antibodies was similar to tumours treated with US+CA plus only CD8+ T cell-depleting antibodies, but both groups had a 3-fold higher tumour volume than US+CA-treated tumours that did not receive T cell depleting antibodies [42].

Wu et al. divided breast tumours (4T1) into the following treatment groups: untreated, US-only, CA-only, and US+CA [43]. They detected a significantly (*p* < 0.05) higher (1.2–1.5-fold) expression of HMGB1 and a significantly (*p* < 0.05) lower ATP level in vitro after US+CA treatment compared to the three other groups and reported an increase in calreticulin expression in the US+CA group in vivo, indicative of DAMP release. A significant (*p* < 0.01 and *p* < 0.05, respectively) increase in IL-12 and Tumour necrosis factor alpha (TNF-α) levels was also reported [43].

In contrast to the other studies, Huang et al. treated lung (LL/2) or colon (CT26) tumour cells with CA only or US+CA and detected no difference in levels of calreticulin, HMGB1, or ATP in vitro 4 h after treatment [44]. There was no significant difference in the percentage of CD80+CD86+ cells (mature DCs) or Interferon gamma (IFN-γ)-secretion in mature DCs after mature DCs were incubated with supernatants from tumour cells treated with US+CA or CA [44].

Overall, several papers confirmed that US+CA can cause DAMP release as well as TAA release. The type of DAMP and amount of DAMP release detected seem to vary between experiments, but this could be due to both different US parameters and treatment protocols. As a consequence of TAA, TSA, and DAMP release, leukocytes may be recruited to tumours and exert their anti-tumour effect [45]. Neither Liu et al. nor Joiner et al. investigated the mechanism underlying DAMP release; however, Hu et al. suggested it was due to tumour cell necrosis, which they observed after treatment.

### 2.2. Modulation of Tumour Vasculature and Perfusion

Tumour vasculature, and the level of perfusion it provides, is important for the presentation of TAAs and TSAs by APCs, and the migration of immune cells such as CTLs into and throughout the tumour [46]. Hence, the impact of US+CA on tumour vasculature and the corresponding immune cell infiltration has been probed in recent papers [33,34,35].

Bulner et al. showed that anti-vascular US-stimulated CAs decreased colon tumour (CT26) growth compared to tumours receiving CAs alone and, perhaps as a consequence of restricted perfusion, observed no significant increase in T cell infiltration [34]. A significant (*p* < 0.05) shutdown of blood flow in tumours followed by necrosis was observed by contrast-enhanced US imaging. Specifically, the peak enhancement intensity was reduced by 88 ± 3.6% relative to CA with no US treatment. Flow cytometry analysis examined the change in CD45+ cells (leukocytes), CD45+CD8+ cells (CTLs), CD45+CD4+ cells (T helper cells), and CD45+CD4+Foxp3+ cells (Tregs) in tumours both 3 and 7 days after the initial treatment day. There was no significant difference between the US+CA group and the CA group. Nonetheless, there was a significant (*p* < 0.05) increase (1.5-fold) in the number of cells in tumour-draining lymph nodes in mice with US+CA-treated tumours 3 days after the initial treatment day, indicating some immune response was established [34].

Contrarily to Bulner et al., Hunt et al. observed an increase in intratumoural immune cells 24 h after anti-vascular US+CA treatment of melanoma tumours (K1735) compared to tumours treated with US-only [35]. Hunt et al. observed a 70% reduction of blood perfusion accompanied by subsequent necrosis. Histopathological analysis suggested that there was a significant (*p* < 0.01) increase in CD45+ cells (leukocytes). More specifically, the number of CD45+ cells per High power field (HPF) was 50 ± 7 in US+CA-treated tumours and 15 ± 1 in US-treated tumours. Likewise, there was a significant (*p* < 0.01) increase in CD3+ cells (T cells) with 60 ± 7 cells/HPF and 11 ± 1 cells/HPF in US+CA-treated and US-treated tumours, respectively. However, there was no difference in B220+ cells (B cells) between the two groups [35].

Similarly, Wu et al. used contrast enhanced US imaging to report significantly (*p* < 0.05) decreased blood perfusion that was not recovered 24 h later [43]. A significant (*p* < 0.01) upregulation of the CD11c+CD80+CD86+ cell (matured DC) proportion in tumour draining lymph nodes was detected after US+CA treatment compared to untreated tumours (26.40 ± 3.70% vs. 13.03 ± 2.03%). In addition, there was a significant (*p* < 0.01) increase in infiltration of CD3+CD8+ cells (CTLs) in tumours after US+CA treatment as measured by flow cytometry [43].

Li et al. looked at the effect of US+CA on tumour blood perfusion and the corresponding immune responses in a preclinical model of colon cancer (MC38) [33]. Mice were injected with CAs and tumours were exposed to a Peak negativepressure (PNP) of either 0.8 MPa or 2.4 MPa. Vasodilation of tumour microvessels and consequently enhanced blood perfusion, as measured by contrast enhanced US, was observed for the 0.8 MPa-treated tumours, while poor blood perfusion, microvascular damage, and hemorrhage in tumour tissues were present in those treated with 2.4 MPa. Further, the percentage of infiltrating CD8+ T cells in tumours 24 h after treatment was 32.35% or 19.43% of the total number of lymphocytes for 0.8 MPa or 2.4 MPa, respectively.

All four studies by Bulner, Hunt, Wu, and Li reported worse blood perfusion after US+CA, although for Li et al. this was only the case for the highest pressure treatment (2.4 MPa). Both Bulner et al. and Hunt et al. assessed and confirmed necrosis—a type of immunogenic cell death that can activate an inflammatory response [47]. Despite this, Bulner et al. did not observe increased tumour infiltration of immune cells. On the other hand, both Hunt et al. and Wu et al. confirmed immune cell infiltration. The use of different US parameters and CAs between studies makes it difficult to compare the reported results, particularly considering that Hunt et al. suggested that thermal changes could play a role. Yet, none of the studies looked into temperature elevation, which could provide valuable information.

Moreover, it has previously been shown that different US parameters promote different anti-vascular effects [48], which could explain the difference in detected immune response. Indeed, tumour perfusion recovery after a 30 min treatment with a PNP of 2.6 MPa and CA has been reported, but with US+CA at a PNP of 4.8 MPa the perfusion was reduced even after 24 h [48]. Hence, it could be that the US parameters used by Hunt et al. and Wu et al. allowed for anti-vascular effects to be reversed within their set time frame, thus explaining the enhanced immune response observed in these tumours despite the initial damage imparted. In contrast, in the studies by Bulner et al. and Li et al., tumour perfusion may not have been restored within their time frame, which may have hindered immune cell trafficking [12]. To gain a better understanding of the presence of tumour-infiltrating immune cells or the lack thereof, it would be valuable for future studies to determine blood perfusion before excising the tumours.

### 2.3. Cellular Effects of Cavitation

US-mediated cavitation may also have a direct or indirect effect on cells important for anti-tumour immunity. These can either be immune cells [36,37] or cells that can suppress immune responses [49].

Lin et al. showed that US+CA could redirect Tumour-associated macrophage (TAM) polarisation and investigated its role in vessel normalisation in a pancreatic tumour model (SW1990) [36]. There are two types of TAMs: M1, which is associated with promoting inflammation and anti-tumour activity, and M2, which typically enhances tumour progression [50]. Lin et al. measured changes in F4/80+ cells (macrophages) by imaging stained tissue (immunohistochemistry) and calculating the percentage of positive F4/80+ area over the total image area. There was a significant (*p* < 0.05) increase in the F4/80+ area after US+CA treatment compared to untreated tumours (29.87 ± 1.16% vs. 20.87 ± 1.17%). Moreover, there was a significant (*p* < 0.001) 2-fold increase in F4/80+CD86+ cells (M1 macrophages) among F4/80+ cells in US+CA-treated tumours compared to untreated tumours, suggesting a shift in TAM polarisation after treatment.

It has previously been demonstrated that shifting from the M2 to the M1 phenotype can normalise tumour vessels and enhance anti-tumour immunity [51]. Indeed, Lin et al. observed, by contrast-enhanced US imaging, a 1.5 fold higher perfusion intensity in the US+CA group than that of the untreated group. To examine the relationship between vessel normalisation and TAM polarisation, they treated tumours with clodronate liposomes to remove TAMs. Firstly, TAM depletion reduced tumour growth in the untreated group, indicating that TAMs were predominantly M2 macrophages. Moreover, the vessel perfusion was improved in TAM-depleted tumours, but there was no significant difference between the untreated and treated group. Thus, it was concluded that the induction of vessel normalisation primarily relied on converting M2-type macrophages to M1-type macrophages [36].

Moreover, Zhang et al. looked at how US+CA may affect VEGF expression and the following immune activation [37]. VEGF can interfere with immune cell migration indirectly by promoting a vascular immune barrier, and more directly by interacting with Vascular endothelial growth factor receptor (VEGFR) on effector T cells, Tregs, DCs, and MDSCs; it can also affect the immune cell phenotype and function [52,53]. Zhang et al. observed that the murine prostate cancer cells (RM-1) exposed to US+CA had significantly (*p* < 0.05) lower VEGF expression, as assessed by western blotting, than those exposed to US-only, CA-only, or untreated cells (32.7 ± 4.9% vs. 47.9 ± 5.9%, 71.7 ± 6.6%, and 74.2 ± 4.7%, respectively). These cells were then co-cultured with CD11c+ DCs and CD8a+ T lymphocytes, and it was demonstrated that there were slight but significant (*p* < 0.05) increases in the amounts of CD11c+ cells (22.2 ± 0.9% vs. 18.1 ± 0.4%) and CD8a+ T cells (34.0 ± 1.5% vs. 31.5 ± 0.8%) when co-cultured with US+CA-treated RM-1 cells than untreated RM-1 cells. Thus, it was concluded that the inhibition of VEGF expression in RM-1 cells caused by US+CA can promote DC and CTL proliferation [37].

**Table 1 pharmaceutics-15-02110-t001:** Summary of treatment protocol, ultrasound (US) parameters, and main findings of the papers mentioned in Section 2.

Author	Treatment Protocol	US Parameters	Main Findings
Liu et al. [40]	CT26 tumours were divided into three treatment groups: (1) untreated, (2) US-only, and (3) US+CA. SonoVue MBs were used as CAs. Bolus of CAs was injected immediately before US exposure. Mice were sacrificed 1, 3, and 18 days after treatment.	*F*^1^ = 0.5 MHz, *PRF* ^2^ = 1 Hz, *PL* ^3^ = 100 ms, and *PNP* ^4^ = 0.6 MPa and 1.4 MPa. Exposure time: 20 s per spot, and 9–12 spots were sonicated to cover the entire tumour. Transducer details: element diameter of 64 mm and radius curvature of 55 mm.	Tumour growth was inhibited for both pressures, albeit it was greater for the higher pressure. The percentage of CD45+CD8+ T cells increased in tumours treated with PNP of 1.4 MPa compared to untreated tumours across all three time points.
Joiner et al. [41]	Mice with KPC tumours were divided into two groups: (1) untreated and (2) US+CA. Lipid MBs with $C_{4}F_{10}$ in the core were used as CAs. CAs were infused for the entire duration of US treatment. Tumours were excised either 2 or 15 days after treatment.	*F* = 1 MHz, *PRF* = 100 Hz, *PL* = 1 ms, and *PNP* = 0.5 MPa. Exposure time: 7– 10 min depending on tumour size. Transducer details: eight-element annular array, 80 mm focus, 1 mm × 1 cm focal spot.	Significant reduction in tumour growth after treatment. The number of CD4+ T cells, CD8+ T cells, and Ly6C-F4/80+CD11b+ macrophages in lymph nodes was significantly higher in treated tumours 2 days after treatment. However, the observed immune response was likely transient due to no significant difference in these immune cell populations being observed 15 days after treatment. A substantial increase in HMGB1 was measured in treated tumours. It was suggested that stable cavitation was the dominant bubble behaviour in this study.
Hu et al. [42]	RM1, MC38, and B19 tumours were divided into four groups: (1) untreated, (2) US+CA, (3) aPD-1, and (4) US+CA+aPD-1. The CA was a lipid nanobubble with $C_{3}F_{8}$ as core gas. US was applied to tumours 5 min after intravenously administering the CAs. aPD-1 was administered intraperitoneally into mice once every 3 days during the treatment period for a total of four doses.	*F* = 1 MHz, *PRF* = 100 Hz, *PL* = 6 ms, and *I* ^5^ = 1 W/cm^2^. Exposure time: 30 s. Transducer details: collimated beam and an effective probe radiation area of 2 cm^2^.	US+CA and US+CA+aPD-1 substantially inhibited tumour growth compared to untreated controls. These two groups had significantly more CD44+CD8+ cells compared to untreated tumours, but the difference was greater for the US+CA+aPD-1 group. There were significantly more granzyme B and IFN-γ-secreting CD8+ T cells in the combination therapy group and the US+CA group compared to the untreated group. They also reported TAA release and DAMP release in vitro when RM-1 cells were treated with US+CA compared to the untreated, US-only, and CA-only groups.
Wu et al. [43]	For the first study, 4T1 tumours were divided into one of the following groups: (1) no treatment, (2) US, (3) CA, or (4) US+CA. Lipid MBs with $C_{3}F_{8}$ as core gas were used as CAs. Tumours were exposed to US immediately after the injection of lipid MBs on days 0, 1, 2, 3, and 4. In another in vivo, they divided the tumour-bearing mice into these four groups: (1) control, (2) aPD-L1, (3) US+CA, and (4) US+CA+aPD-L1. The treatment strategy was similar to the first in vivo, but aPD-L1 was injected intravenously on days 1, 4, and 7. Mice were sacrificed on day 11 after treatment.	*F* = 1 MHz, *PRF* = not specified, *PL* = not specified, *I* = 3.0 W/cm^2^, and duty cycle = 50%. Exposure time: 5 min. Transducer details: diameter of 1 cm, focal length of 1.5 cm, and focus area of 0.4 cm^2^.	In the first study, a substantial difference was reported in tumour volume after treating with US+CA compared to the untreated, US-only, and CA-only groups. The tumour blood perfusion was blocked even 24 h after treatment. There was a significant increase in CD11c+CD80+CD86+ cells (matured DCs) and CD3+CD8+ T cells, as well as the level of IL-12 and TNF-α cytokines. In the second study, it was observed that the tumour growth was even more inhibited by the US+CA+aPD-L1 and there was a remarkable increase in activated CD8+ T cell infiltration compared to untreated tumours and aPD-L1-treated tumours.
Huang et al. [44]	LL/2 and CT26 tumour cells and tumours were treated with either (1) CA or (2) US+CA. The CA was a lipid MB with $C_{3}F_{8}$ in the core. CAs with or without US exposure were administered every 3 days, for a total of 6 treatments (18 days), and mice were observed for 28 days. US exposure occurred 1 min after CA administration.	*F* = 2.25 MHz, *PRF* = 1 Hz, *PL* = 10 ms, and *PNP* = 1.9 MPa. Exposure time: 10 min. Transducer details: diameter of 20 mm and focal length of 50 mm.	Tumour cells treated with only CA or US+CA did not show induced translocation of calreticulin or Erp57, or release of HMGB1 or ATP in vitro. There was no substantial difference in CD80+CD86+ cells (mature DCs) or IFN-γ-secreting cells in vivo between the two groups.
Bulner et al. [34]	Mice with CT26 tumours were randomised into these groups: (1) CA, (2) aPD-1, (3) US+CA, and (4) US+CA+aPD-1. The CAs were MBs consisting of lipids encapsulating $C_{3}F_{8}$ gas. US exposure commenced immediately after intravenous injection of CAs. For the acute experiments, animals sacrificed at day 3 received US+CA or CA treatment at day 3 with or without aPD-1 at day 0. Animals sacrificed at day 7 received US+CA or CA treatment with or without aPD-1 at day 0, 3, and 6. For the longitudinal experiment, the treatment schedule was similar but aPD-1 was administered on day 9 and 12 as well, and mice were sacrificed at day 30.	*F* = 1 MHz, *PRF* = 100 Hz, *PL* = 0.1 ms, and *PNP* = 1.65 MPa. Exposure time: 50 pulses were repeated at 20 s interval for a duration of 2 min. Transducer details: spherically focused, 3.75 cm diameter, 15 cm focal length, and 1.05 cm −6 dB beam width.	The improved tumour growth inhibition was attributed to the shutdown of blood flow due to no evidence supporting a T cell-dependent mechanism (CD45+CD8+ cells and CD45+CD4+ cells). However, the re-challenge experiment suggested an engagement of adaptive memory response. Passive cavitation detection detected broadband noise.
Hunt et al. [35]	Animals with K1735 tumours were initially divided into three treatment groups: (1) 3 min US exposure, (2) US+CA with 1 min exposure, and (3) US+CA with 3 min exposure. Definity MBs were used as CAs. Tumours were insonated immediately after intravenous injection of CAs. Mice were sacrificed 24 h after treatment.	*F* = 3 MHz, *PRF* = continuous, *PL* = continuous, *PNP* = 0.22 MPa, and exposure time: 1 min or 3 min. Transducer details: unfocused, power level 3, and spatial average intensity 2.3 W/cm^2^.	A significant shutdown of blood flow after both US+CA treatments compared to US-treated tumours was reported. There was a significant increase in the mean number of CD45+ cells and CD3+ cells after US+CA ( 3 min) compared to untreated tumours.
Li et al. [33]	MC38 tumour-bearing mice were divided into the following groups for the perfusion study: (1) US+CA at 0.8 MPa, (2) US+CA at 2.4 MPa, (3) US at 2.4 MPa, and (4) untreated control. For the second study, the groups were: (1) untreated, (2) US+CA at 0.8 MPa, (3) aPD-L1, and (4) US+CA+aPD-L1 at 0.8 MPa. Sonazoid MBs were used as CAs. The CAs were slowly injected through tail vein during US exposure. For the perfusion study, mice were sacrificed 24 h after treatment. For the combination therapy, the mice were injected with aPD-L1 on days 4, 7, 10, and 13, US+CA treatment was performed 24 h after each aPD-L1 administration, and mice were sacrificed 24 h after the final treatment.	*F* = 4 MHz, *PRF* = 1 kHz, *PL* = 4.5 μs, and *PNP* = 0.8 MPa and 2.4 MPa. Exposure time: 1 s on and 1 s off for 10 min. Transducer details: phased focus.	The 0.8 MPa exposure enhanced perfusion substantially, whereas the 2.4 MPa reduced blood perfusion. There was a significant increase in CD8+ T cells for the 0.8 MPa exposure compared to the untreated tumours and US+CA treatment at 2.4 MPa. Moreover, US+CA+aPD-L1 had significantly better therapeutic effect and more CD8+ T cells than US+CA and aPD-L1 only. Additionally, the US+CA+aPD-L1 boosted IFN-γ and granzyme B secretion.
Lin et al. [36]	Mice with SW1990 tumours were randomised into (1) untreated and (2) US+CA groups. MBs from Bracco were used as CAs. Tumours were treated 5 days per week. The CAs were slowly injected via the tail vein.	*F* = 1 MHz, *PRF* = 1 kHz, *PL* = 0.2 ms, and *I* = 1.2 W/cm^2^. Exposure time: 2 min five times. Transducer details: not specified.	It was demonstrated that the induction of vessel normalisation by US+CA mainly relied on shifting TAM polarisation from M2-type to M1-type.
Zhang et al. [37]	For the VEGF expression experiment, RM-1 cells were divided into the following groups: (1) US, (2) CA, (3) US+CA, and (4) untreated. The CAs were SonoVue MBs. To detect DC and T lymphocyte phenotype, RM-1 cells either treated with US+CA or untreated were co-cultured with DCs and T lymphocytes.	*F* = 800 kHz, *PRF* = 1 Hz, *PL* = 0.5 s, and *ISATA* ^6^ = 360 mW/cm^2^. Exposure time: 30 s. Transducer details: cylindrical probe with a diameter of 13 mm.	VEGF expression was significantly decreased after US+CA treatment compared to the other groups. There was a significant increase in CD11c+ DCs and CD8a+ T cells in the US+CA group compared to the untreated group.
Tan et al. [49]	Spleens of LLC tumour-bearing mice were in the first study divided into two groups: (1) US+CA and (2) untreated. The CAs were Sonazoid MBs. Diluted CA of 0.1 mL was injected at the first 100 s, 0.02 mL per 100 s for three times, and 0.02 mL per 50 s at the rest of the treatment. Tumours and spleens were excised 24 h after treatment. In the second study, mice were divided into the following groups: (1) US+CA, (2) aPD-L1, (3) US+CA+aPD-L1, and (4) untreated. Spleens were treated once every 3 days for a total of 3 times with US+CA, and aPD-L1 was injected intraperitoneally on the following day. Spleens and tumours were excised on day 12.	*F* = 5 MHz, *PRF* = 500 Hz, *PL* = 1.3 μs, *PNP* = 2.2 MPa, and 2.3 MPa. Exposure time: transmitting and intermittent time of 0.1 s, and total duration 600 s. Transducer details: linear array probe.	In the first study, the results showed a significant reduction in splenic CECs and an increase in CD8+ T cells when treated with US+CA compared to untreated spleens. There was no substantial difference in CD11b+Gr1+ cells (MDSCs), CD11b+CD11c+ cells (DCs), CD11b+F4/80+ cells (macrophages), or B220+ cells (B cells). In the second study, tumour growth was only inhibited when US+CA was combined with aPD-L1. The US+CA+aPD-L1 treatment demonstrated a significant increase in number of IFN-γ-producing CD8+ T cells and CD4+ T cells.

^1^ Frequency, ^2^ pulse repetition frequency, ^3^ pulse length, ^4^ peak negative pressure, ^5^ intensity, and ^6^ spatial average time average intensity.

In another study, Tan et al. exposed spleens rather than tumours to US+CA in a lewis lung cancer model [49]. The reason for this was that immature red blood cells called CD71+ erythroid progenitor cells (CECs), which expand in the spleen, suppress immune responses [54,55]. Immunosuppression can induce tumour immune evasion to promote tumour growth. Additionally, CECs also modulate T cells via the PD-L1/PD-1 pathway and express genes encoding immune checkpoint molecules [56]. Thus, by targeting and consequently reducing the CECs residing in the spleen, the anti-tumour immune response might be enhanced. Indeed, Tan et al. detected a significant (*p* = 0.006) decrease in CD71+TER119+ cells in the spleen, and the percentage of CD8+ T cells was significantly (*p* = 0.003) increased after US+CA treatment compared to untreated spleens. However, there was no difference in tumour growth between these two groups. Likewise, there was no difference in the number of CD11b+Gr1+ cells (MDSCs), CD11b+CD11c+ cells (DCs), CD11b+F4/80+ cells (macrophages), or B220+ cells (B cells) in the spleen [49]. It is unclear how US+CA would impact specifically on the intended cell targets within the spleen without also disrupting the general immune cell profile.

Altogether, US-mediated cavitation can contribute to the immunomodulation of tumours through a variety of mechanisms: releasing DAMPs and TAAs, altering tumour perfusion, and modifying cellular activity. These changes to the tumour immune microenvironment might produce a TME more favourable for ICI therapies.

## 3. Cavitation-Enhanced Checkpoint Inhibitor Therapy

In addition to the mounting evidence that US-mediated cavitation can promote an anti-tumour immune response without drug addition, there is also gathering evidence that combining US+CA with immunotherapeutic drugs may provide an additional benefit to the treatment. Cancer immunotherapy, particularly the use of ICIs, is considered to be one of the most promising strategies to treat cancer [57], and thus combining US-mediated cavitation with ICIs could be a powerful treatment strategy. Cavitation has the potential to both improve the drug delivery of ICIs and, as previously discussed, engage the immune system.

### 3.1. Improved Delivery

A drawback of ICIs is their relatively large size (∼ 150 kDa), which limits their intratumoural delivery [58] and consequently their therapeutic efficacy [59]. US-mediated cavitation has been shown to improve the intratumoural delivery of drugs [60]. In fact, Grundy et al. observed a 2.1–3.6-fold increase in cetuximab concentration in murine tumours targeted with US+CA compared to CA-treated tumours [61]. Moreover, Li et al. reported a significant (*p* < 0.0001) 1.37-fold increase in concentration of the aPD-L1 antibody in the tumour when co-administered with US+CA compared to aPD-L1 monotherapy [33]. In addition, Kim et al. measured a significant (*p* < 0.01) increased intratumoural staining of aPD-L1 after administering aPD-L1-coated CA and exposing to US compared to aPD-L1-coated CA without US, free aPD-L1 monotherapy, or US+aPD-L1 without CA [62]. Thus, there are data showing that US+CA can increase the tumour uptake of therapeutic antibodies. Even though the poor response to ICI in some patients might be due to poor delivery, the heterogeneous tumour immune environment is also an important consideration for ICI efficacy.

### 3.2. Favourable Tumour Immune Microenvironment

It has been reported that having a sufficient CD8+ T cell presence within the TME favours tumour susceptibility to PD-1-based therapy [63]. In other words, “hot” tumours, defined as highly T cell-infiltrated tumours, are associated with a better treatment response. In contrast, “cold” tumours that typically have poor T cell infiltration limit the efficacy of ICIs [64]. Converting a “cold” tumour into a “hot” tumour and successfully activating T cells could be essential for sufficient therapeutic effect. As discussed in Section 2, US-mediated cavitation can contribute to making the tumour immune microenvironment more favourable for ICIs.

Indeed, Li et al. observed a significant (*p* < 0.05 and *p* < 0.001, respectively) increase in tumour growth inhibition when treated with US+CA+aPD-L1 compared to US+CA or aPD-L1 alone. There was a significant (*p* < 0.05) increase in the percentage of CD8+ T cells, as well as a significant (*p* < 0.01) increase in IFN-γ and granzyme B secretion in tumours receiving US+CA+aPD-L1 [33]. Similarly, Hu et al. reported substantial tumour growth inhibition in three different tumour models when tumours were treated with US+CA+aPD-1 compared to US+CA or aPD-1 alone [42]. All tumour models were characteristically “cold” tumours, but after US+CA+aPD-1 treatment they observed significantly (*p* < 0.05) more CD44+CD8+ cells and the presence of T cell effector functions compared to monotherapies. The US+CA+aPD-1 group also exhibited a high expression of genes related to ’immune response’, ’T cell activation’, and ’antigen processing and presentation’ [42]. Wu et al. reported greater tumour growth inhibiton in US+CA+aPD-L1-treated tumours than in the US+CA group and aPD-L1 group, and the combination therapy resulted in a substantial increase in activated CD8+ T cell infiltration [43].

Tan et al. treated spleens with US+CA+aPD-L1 and showed that tumour volume was significantly (*p* < 0.001) reduced compared to tumours in the US+CA, aPD-L1 alone, or untreated groups [49]. The results demonstrated that the number of IFN-γ-producing CD8+ T cells and CD4+ T cells significantly (*p* = 0.0134 and *p* = 0.0082, respectively) increased in US+CA+aPD-L1-treated spleens compared to untreated spleens. Tan et al. hypothesised that US+CA prevented CECs from inhibiting CD8+ T cell proliferation in the spleen, so more cytotoxic T cells in the peripheral circulation system could benefit from aPD-L1 [49]. Notably, although the spleen was located by a small animal US system, cavitation was not mapped or quantified to ensure that cavitation was confined to the spleen alone.

Bulner et al. also observed enhanced tumour growth inhibition for mice with tumours treated with US+CA+aPD-1 compared to both the US+CA group and aPD-1-only group, but a clear T cell-dependent mechanism was not detected [34]. However, there was a significant (*p* < 0.05) increase in the number of cells in the tumour draining lymph node 7 days after US+CA+aPD-L1 treatment compared to US+CA treatment but not aPD-1 treatment. The cells within the lymph node were not characterised. Additionally, one mouse that exhibited complete regression was subjected to a re-challenge experiment with the same tumour cell line, and no tumour was present on US imaging after 90 days, suggesting an adaptive immune response prevented tumour growth upon re-challenge. Hence, a potential contribution of anti-tumour immune responses should not be excluded.

### 3.3. Reduced Adverse Effects

Lastly, US and CAs may limit the impact of drug in non-target tissue as well as reducing Immune-related adverse effects (irAEs). An increasing number of studies on irAEs are being reported with approximately 10–20% of patients treated with aPD-L1 showing irAEs [62,65]. Kim et al. showed that US and CAs coated with aPD-L1 significantly (*p* < 0.05) reduced tumour volume compared to US+aPD-L1 without CA, aPD-L1-coated CA without US, or free aPD-L1. Moreover, it was reported that 18/20 mice that received the aPD-L1-coated CA without US survived 15 days post-treatment, whereas 9/20 mice survived after administration of free aPD-L1 [62]. Hence, incorporating ICIs into CAs may minimise ICI toxicities.

There are multiple ways in which US-mediated cavitation can improve ICI therapy. Although US+CA has the potential to improve the delivery of ICIs, the efficacy of ICIs is not necessarily improved. Similarly, though reducing the adverse effects of ICIs is important, enhancing the therapeutic efficacy is still desirable. Thus, if the aim is to enhance the anti-tumour effect of ICIs, converting “cold” tumours into “hot” tumours by US-mediated cavitation is an attractive strategy.

## 4. Discussion

It is clear from the reported data that US-mediated cavitation can affect anti-tumour immunity in several ways, even without the administration of additional therapeutics, although the responses are mostly transient. Furthermore, evidence suggests that cavitation can contribute to enhancing the efficacy of ICIs. However, gaining a better understanding of how cavitation increases the therapeutic effect of ICIs is essential for both improving upon and developing new treatment approaches.

With the sparse data available, there are likely still multiple immunological effects of cavitation that remain undefined. Thus, to achieve the most optimal effect of US-mediated cavitation in combination with ICIs, it would be valuable to further explore what immunological effects cavitation may induce. Previous studies have begun to profile the negative and positive effects of cytokines and chemokines on prognosis following ICI treatment [66,67], and there are some studies indicating that cavitation can affect cytokine presence [43]. Defining the impact of the type and duration of cavitation on the release of chemokines and cytokines could help link cavitation outputs to potential ICI efficacy. Similarly, assessing the effect of cavitation on cancer-associated fibroblasts or NK cells could be interesting since they both have important roles in anti-tumour immunity [68,69].

More importantly, whether there is a direct relationship between the oscillation regime and the biological and immunological effects remains to be elucidated. It is currently unclear whether the gentle oscillation or violent collapse of bubbles is more favourable to achieve certain immunological effects like DAMP release, immune cell migration, and change in cellular activity such as TAM polarisation. It is suggested that the microjets that arise from the violent collapse of bubbles can cause irreversible cell damage and necrotic cell death [70], which would consequently result in the release of DAMPs and TAAs. However, stressed cells are also known to release DAMPs [71], and thus shear stress generated by microstreaming from oscillating bubbles can also explain the observed DAMP release. Furthermore, Matsuura reported the detection of broadband noise while observing a reduction in perfusion [72], suggesting that violent collapse might be responsible for the shutdown of blood flow. However, the type of oscillation regime has not been confirmed for the enhanced tumour perfusion observed by Li et al. [33].

In regards to tumour vasculature, there are advantages and disadvantages to both vascular effects caused by cavitation. First, anti-vascular effects can cause tissue necrosis to activate immune responses and produce numerous antigens to induce APC maturation [73]. However, vascular disruption will also restrict the access of immune cells and therapeutic drugs. Thus, it is important for future studies to determine the long-term vascular effects of cavitation and assess blood flow immediately before tumour excision. Additionally, it is important to consider the sequencing and timing of US exposure and drug administration. On the other hand, cavitation that causes enhanced perfusion has the disadvantage of potentially not eliciting a strong immune response due to lack of DAMP and TAA release. In these situations, it is useful to know which bubble behaviour causes which vascular effect to ensure that the desired effect is achieved.

Notably, bubble behaviour is not only dependent on US parameters such as frequency, PNP, pulse repetition frequency, and sonication duration but also the characteristics of the CAs, such as their size, composition, and concentration [74]. Additionally, the tumour type, location, and environment will affect the bubble oscillation and the subsequent physical mechanisms. Thus, passive cavitation detection, i.e., monitoring cavitation behaviour, is an essential tool in terms of uncovering which bubble behaviour is dominant and desirable for certain biological and immunological effects [32,75,76,77]. Yet, only one paper in Section 2 assessed cavitation levels, allowing Bulner et al. to report which acoustic emissions were dominant [34], although Joiner et al. confirmed the presence of CAs in tumours after treatment by contrast enhanced imaging. Thus, it would be beneficial if future studies included passive cavitation detection to provide more details about the oscillation regime.

Moreover, a better understanding of the level, type, and intratumoural location of cavitation events will not only be of vital importance in optimising the efficacy of this approach but also its safety as it progresses to the clinic. Indeed, technology that allows a clinician to verify and map the level and type of cavitation taking place during treatment of a patient with US+CA+ICI will be of huge benefit. However, it is noteworthy that the recorded frequency content should be interpreted with care: although the violent oscillation of bubbles (inertial cavitation) emits broadband signals, the range of bubble sizes is continually changing, and this can also produce acoustic emissions with a wide range of frequencies [78]. Moreover, harmonic frequencies may be produced both by non-linear propagation and oscillating bubbles [79].

## 5. Conclusions

This review attempts to give an overview of some of the biological effects that cavitation may induce to promote an anti-tumour immune response and how cavitation may be combined with checkpoint inhibitors. The aforementioned papers suggest different ways US-mediated cavitation can enhance immune cell presence and activity in tumours, including DAMP and tumour antigen release, vascular modification, and the alteration of cellular activity. However, due to the limited number of studies and each of them either using different tumour models and tumour stages, treatment protocol, immune cell markers, or US parameters, it is difficult to draw any specific conclusions on how cavitation may reverse immune escape.

Compiling additional reproducible studies would be beneficial and improve the understanding of cavitation-mediated immunomodulation. Moreover, most studies do not report which oscillation regime is dominant or fully define the level and location of cavitation. By gaining a better understanding of which physical mechanisms trigger the desired anti-tumour immune responses, it may be easier to choose US parameters that ICIs and other immunotherapeutic drugs will benefit from the most and help achieve consistent results. A particular emphasis on how cavitation may turn “cold” tumours “hot” and thereby widen the clinical response to ICIs will be key.

## Figures and Tables

**Figure 1 pharmaceutics-15-02110-f001:**
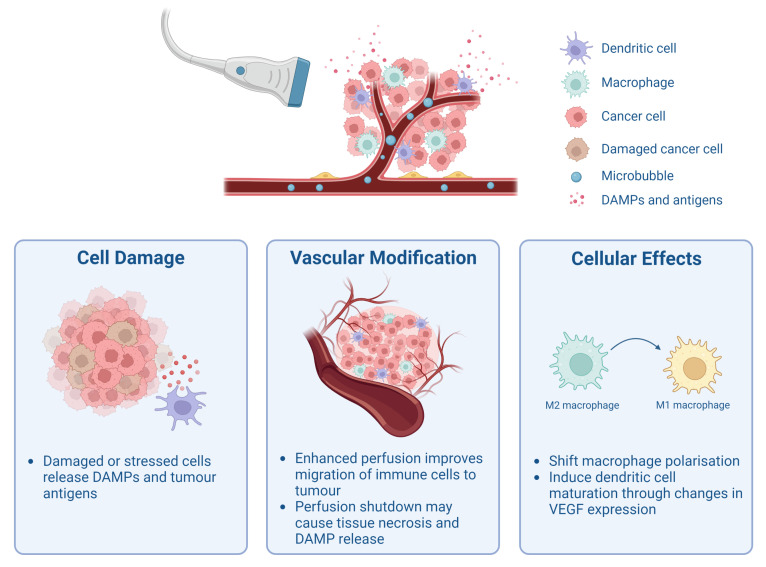
Schematic overview of how ultrasound-mediated cavitation may affect different parts of the tumour. It has been proposed that cavitation can cause cell damage and stress, which consequently releases Damage-associated molecular patterns (DAMPs) and tumour antigens. Moreover, it has been suggested that depending on the type, duration, and intensity of cavitation, tumour perfusion can either be increased [33] or blood flow can be shut down [34,35]. Either situation may lead to increased anti-tumour immunity as a result of improved migration and infiltration of immune effector molecules and cells or due to increased tissue necrosis and Damage-associated molecular pattern (DAMP) release. Lastly, ultrasound-mediated cavitation has been shown to transform M2 macrophages to M1 macrophages [36] and additionally induce dendritic cell maturation by altering expression of Vascular endothelial growth factor (VEGF) [37]. Figure created with BioRender.

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
