# Peer review of "Cavitation-Mediated Immunomodulation and Its Use with Checkpoint Inhibitors"

_pharmaceutics, 2023, doi:10.3390/pharmaceutics15082110_

Round 1

Reviewer 1 Report

In this manuscript, Maardalen et al. reviewed the current evidence regarding the cavitation-mediated immunomodulation and its use with checkpoint inhibitors, and discuss the challenges and difficulties in the management of these patients. The authors considered that cavitation may be closely related to immunomodulation and potentially serve as a method to enhance the delivery and activity of immune checkpoint inhibitors. This review is important for promote anti-tumours immune responses by cavitation in various cancer. There are several concerns with the study:

1. The second section describes the mechanisms of Cavitation-Mediated immunomodulation in detail, whereas the evidence for immunotherapy in various cancer is limited. The authors should add more relevant literatures to improve the credibility.

2. The authors should discuss more about the future expectations of cavitation in checkpoint inhibitors treatment.

3. Some important studies regarding the immunotherapy and cavitation should be cited and discussed. For example, PMID: 32165014.

4. A summarized/graphical figure should be added to help readers understand this review clearly.

5. There are still some grammatical and spelling errors throughout.

 Moderate editing of English language required

Author Response

  1. The second section describes the mechanisms of Cavitation-Mediated immunomodulation in detail, whereas the evidence for immunotherapy in various cancer is limited. The authors should add more relevant literatures to improve the credibility.

    We agree that the evidence of the utility of immunotherapy could have been better emphasised and so the last paragraph of section 1 (lines 91-107) have been adapted as below:

    In summary, tumours have multiple methods to evade immune recognition. Con- 91
    sequently, there are several ways in which an anti-tumour immune response may be 92
    reactivated. Indeed, immunotherapy is a modality that includes a range of treatment 93
    methods, including (1) cancer vaccines which aim to activate CTLs[16], (2) CAR T cells 94
    which are patient derived T cells with a chimeric antigen receptor that are engineered to 95
    recognise and bind specific antigens on the surface of cancer cells[17], and (3) Immune 96
    Checkpoint Inhibitors (ICIs) which interact with immune checkpoints thereby preventing 97
    immune tolerance[18]. The effectiveness of these treatments have been proven in a range 98
    of cancers[19–21]. In fact, the use of ICIs demonstrated that 12 out of 12 patients with rectal 99
    cancer had a clinical complete response[22]. It is notable that the greatest ICI successes 100
    have come in patient populations that have been carefully selected for an immune “hot” 101
    rather than “cold” phenotype[22]. For more widespread impact, methods to optimise such 102
    approaches are needed, and driving tumours towards a “hot” phenotype is thought to be 103
    an effective way of achieving this. Recent research suggests that Ultrasound (US)-mediated 104
    cavitation can result in immunomodulation of the TME, thus partly reversing tumour 105
    immune evasion. Hence, this may be a promising strategy to promote tumour-specific 106
    immune responses and enhance efficacy of ICIs. 107

  2. The authors should discuss more about the future expectations of cavitation in checkpoint inhibitors treatment.

    The discussion now includes more content regarding the studies needed to in order to enhance our understanding of how cavitation may benefit or restrict checkpoint inhibitor treatment, which is important when improving upon or developing new cavitation+checkpoint inhibitor treatment approaches. Lines 430-439 specifically discuss future studies for cavitation and checkpoint inhibitors, whereas the other paragraphs (lines 440-484) are more indirectly related to checkpoint inhibitors. 

    With the sparse data available, there are likely still multiple immunological effects 430
    of cavitation that remain undefined. Thus, to achieve the most optimal effect of US- 431
    mediated cavitation in combination with ICIs it would be valuable to further explore what 432
    immunological effects cavitation may induce. Previous studies have begun to profile the 433
    negative and positive effects of cytokines and chemokines on prognosis following ICI 434
    treatment[66,67], and there are some studies indicating that cavitation can affect cytokine 435
    presence[43]. Defining the impact of the type and duration of cavitation on the release 436
    of chemokines and cytokines could help link cavitation outputs to potential ICI efficacy. 437
    Similarly, assessing the effect of cavitation on cancer-associated fibroblasts and/or NK cells 438
    could be interesting, since they both have important roles in anti-tumour immunity[68,69]. 439

    More importantly, whether there is a direct relationship between the oscillation regime 440
    and the biological and immunological effects remains to be elucidated. It is currently 441
    unclear whether gentle oscillation or violent collapse of bubbles is more favourable to 442
    achieve certain immunological effects like DAMP release, immune cell migration, and 443
    change in cellular activity such as TAM polarisation. It is suggested that the microjets that 444
    arise from violent collapse of bubbles can cause irreversible cell damage and necrotic cell 445
    death[70], which would consequently result in release of DAMPs and TAAs. However, 446
    stressed cells are also known to release DAMPs[71] and thus shear stress generated by 447
    microstreaming from oscillating bubbles can also explain the observed DAMP release. 448
    Furthermore, Matsuura reported detection of broadband noise while observing a reduction 449
    in perfusion[72], suggesting that violent collapse might be responsible for the shutdown 450
    of blood flow. However, the type of oscillation regime has not been confirmed for the 451
    enhanced tumour perfusion observed by Li et al[33]. 452

    In regards to tumour vasculature, there are advantages and disadvantages to both 453
    vascular effects caused by cavitation. First, anti-vascular effects can cause tissue necrosis to 454
    activate immune responses and produce numerous antigens to induce APC maturation[73]. 455
    However, vascular disruption will also restrict access of immune cells and therapeutic 456
    drugs. Thus, it is important for future studies to determine the long-term vascular effects 457
    of cavitation and assess blood flow immediately before tumour excision. Additionally, it is 458
    important to consider the sequencing and timing of US exposure and drug administration. 459
    On the other hand, cavitation that causes enhanced perfusion has the disadvantage of 460
    potentially not eliciting a strong immune response due to lack of DAMP and TAA release. 461
    In these situations it is useful to know which bubble behaviour causes which vascular effect 462
    to ensure that the desired effect is achieved. 463

    Notably, bubble behaviour is not only dependent on US parameters such as frequency, 464
    PNP, pulse repetition frequency, and sonication duration, but also the characteristics of the 465
    CAs, such as their size, composition, and concentration[74]. Additionally, the tumour type, 466
    location, and environment will affect the bubble oscillation and the subsequent physical 467
    mechanisms. Thus, passive cavitation detection, i.e. monitoring cavitation behaviour, is an 468
    essential tool in terms of uncovering which bubble behaviour is dominant and desirable 469
    for certain biological and immunological effects[32,75–77]. Yet, only one paper in section 2 470
    assessed cavitation levels, allowing Bulner et al. to report which acoustic emissions were 471
    dominant[34], although Joiner et al. did confirm the presence of CAs in tumours after 472
    treatment by contrast enhanced imaging. Thus, it would be beneficial if future studies 473
    included passive cavitation detection to give more details about the oscillation regime. 474

    Moreover, a better understanding of the level, type, and intratumoural location of 475
    cavitation events will not only be of vital importance in optimising the efficacy of this 476
    approach, but also its safety as it progresses to the clinic. Indeed, technology which allows 477
    a clinician to verify and map the level and type of cavitation taking place during treatment 478
    of a patient with US+CA+ICI will be of huge benefit. However, it is noteworthy that the 479
    recorded frequency content should be interpreted with care: although violent oscillation of 480
    bubbles (inertial cavitation) emits broadband signals, the range of bubble sizes is continually 481
    changing and this can also produce acoustic emissions with a wide range of frequencies[78]. 482
    Moreover, harmonic frequencies may be produced both by non-linear propagation and 483
    oscillating bubbles[79]. 484

  3. Some important studies regarding the immunotherapy and cavitation should be cited and discussed. For example,PMID:32165014.

    The suggested paper is a well written review on cavitation nuclei for therapy and drug delivery. It mentions physical/mechanical phenomena that may arise from bubble oscillation and is therefore cited (ref 32) when talking about microstreaming and microjets. 

    First, gentle oscillation 122
    of CAs will cause momentum transfer from the bubble to the surrounding liquid, called 123
    microstreaming[31]. This can result in shear stresses impacting on cell membranes[32]. 124
    More violent oscillation and collapse of CAs near a surface can produce shock waves 125
    and microjets, leading to cellular injury and tissue destruction[31,32].

  4. A summarized/graphical figure should be added to help readers understand this review clearly.

    A figure that summarises the immunological effects of cavitation have been added to the paper (Figure 1, see attachment). 

  5. There are still some grammatical and spelling errors throughout.

    The manuscript has been re-checked and errors identified and removed.

Reviewer 2 Report

Overall, this paper provides a comprehensive assessment of the potential impact of cavitation on anti-tumour immune responses. However, there are several areas in which the paper could be improved:

1. While the paper briefly mentions the importance of understanding the biological and physical mechanisms underlying cavitation's promotion of anti-tumour immune responses, it falls short in providing a thorough analysis of these mechanisms. Expanding on this aspect would provide more insights and contribute to the scientific understanding of the topic.

2. The paper could benefit from a clear outline of the limitations and challenges associated with the application of ultrasound-mediated cavitation for enhancing immune checkpoint inhibitors. Addressing these limitations would help guide future research directions and foster a more balanced discussion.

3. It would be helpful to discuss potential future directions for research in this area. Suggesting novel experimental approaches or highlighting unanswered questions would encourage further investigation and advancement of the field.

4. If the author inserts schematic diagrams of typical examples in the article, it will greatly help readers understand.

Minor editing of English language required

Author Response

  1. While the paper briefly mentions the importance of understanding the biological and physical mechanisms underlying cavitation's promotion of anti-tumour immune responses, it falls short in providing a thorough analysis of these mechanisms. Expanding on this aspect would provide more insights and contribute to the scientific understanding of the topic.

    We have included a new paragraph which introduced the mechanical effects of acoustic cavitation in lines 121-126. Moreover, lines 440-452 aim to discuss which underlying mechanical effect/oscillation regime could be responsible for the immunological effects. 

    US can also be used to induce non-ablative mechanical effects to tumours. Acoustic 116
    cavitation is used to describe the expansion and collapse of gas bubbles in response to the 117
    alternating compressional and rarefactional US wave. Notably, providing agents to nucleate 118
    such acoustic cavitation, i.e. by adding pre-existing gas bubbles - referred to henceforth as 119
    Cavitation Agent (CA), can lower the pressure amplitude required to achieve cavitation and 120
    consequently reduce the energy deposition within tissue. The volumetric changes can lead 121
    to various mechanical phenomena that can impact the surroundings. First, gentle oscillation 122
    of CAs will cause momentum transfer from the bubble to the surrounding liquid, called 123
    microstreaming[31]. This can result in shear stresses impacting on cell membranes[32]. 124
    More violent oscillation and collapse of CAs near a surface can produce shock waves 125
    and microjets, leading to cellular injury and tissue destruction[31,32]. The scale of the 126
    membrane disruption and damage, and the underlying mutations to apoptotic pathways 127
    within the cells will determine the consequent route to death (apoptosis or necrosis) and 128
    the resulting immune response. 129

    More importantly, whether there is a direct relationship between the oscillation regime 440
    and the biological and immunological effects remains to be elucidated. It is currently 441
    unclear whether gentle oscillation or violent collapse of bubbles is more favourable to 442
    achieve certain immunological effects like DAMP release, immune cell migration, and 443
    change in cellular activity such as TAM polarisation. It is suggested that the microjets that 444
    arise from violent collapse of bubbles can cause irreversible cell damage and necrotic cell 445
    death[70], which would consequently result in release of DAMPs and TAAs. However, 446
    stressed cells are also known to release DAMPs[71] and thus shear stress generated by 447
    microstreaming from oscillating bubbles can also explain the observed DAMP release. 448
    Furthermore, Matsuura reported detection of broadband noise while observing a reduction 449
    in perfusion[72], suggesting that violent collapse might be responsible for the shutdown 450
    of blood flow. However, the type of oscillation regime has not been confirmed for the 451
    enhanced tumour perfusion observed by Li et al[33].

  2. The paper could benefit from a clear outline of the limitations and challenges associated with the application of ultrasound-mediated cavitation for enhancing immune checkpoint inhibitors. Addressing these limitations would help guide future research directions and foster a more balanced discussion.

    The discussion aims to outline challenges with ultrasound-mediated cavitation for enhancing checkpoint inhibitors, and each paragraph mentions a challenge/limitation. One paragraph (lines 430-439) discuss how the sparse data prevents us from fully understanding the immune effects and more studies should therefore be carried out. Another paragraph (lines 440-452) discuss how there are limited evidence as to which bubble behaviour causes which immune effect. The fourth paragraph in the discussion (lines 453-462) discuss how knowing whether the cavitation enhances or shuts down tumour perfusion is important regarding the timing of drug administration and US exposure. The last two paragraphs of the discussion (lines 464-484) discuss how cavitation detection will be important for future studies. 

    With the sparse data available, there are likely still multiple immunological effects 430
    of cavitation that remain undefined. Thus, to achieve the most optimal effect of US- 431
    mediated cavitation in combination with ICIs it would be valuable to further explore what 432
    immunological effects cavitation may induce. Previous studies have begun to profile the 433
    negative and positive effects of cytokines and chemokines on prognosis following ICI 434
    treatment[66,67], and there are some studies indicating that cavitation can affect cytokine 435
    presence[43]. Defining the impact of the type and duration of cavitation on the release 436
    of chemokines and cytokines could help link cavitation outputs to potential ICI efficacy. 437
    Similarly, assessing the effect of cavitation on cancer-associated fibroblasts and/or NK cells 438
    could be interesting, since they both have important roles in anti-tumour immunity[68,69]. 439

    More importantly, whether there is a direct relationship between the oscillation regime 440
    and the biological and immunological effects remains to be elucidated. It is currently 441
    unclear whether gentle oscillation or violent collapse of bubbles is more favourable to 442
    achieve certain immunological effects like DAMP release, immune cell migration, and 443
    change in cellular activity such as TAM polarisation. It is suggested that the microjets that 444
    arise from violent collapse of bubbles can cause irreversible cell damage and necrotic cell 445
    death[70], which would consequently result in release of DAMPs and TAAs. However, 446
    stressed cells are also known to release DAMPs[71] and thus shear stress generated by 447
    microstreaming from oscillating bubbles can also explain the observed DAMP release. 448
    Furthermore, Matsuura reported detection of broadband noise while observing a reduction 449
    in perfusion[72], suggesting that violent collapse might be responsible for the shutdown 450
    of blood flow. However, the type of oscillation regime has not been confirmed for the 451
    enhanced tumour perfusion observed by Li et al[33]. 452

    In regards to tumour vasculature, there are advantages and disadvantages to both 453
    vascular effects caused by cavitation. First, anti-vascular effects can cause tissue necrosis to 454
    activate immune responses and produce numerous antigens to induce APC maturation[73]. 455
    However, vascular disruption will also restrict access of immune cells and therapeutic 456
    drugs. Thus, it is important for future studies to determine the long-term vascular effects 457
    of cavitation and assess blood flow immediately before tumour excision. Additionally, it is 458
    important to consider the sequencing and timing of US exposure and drug administration. 459
    On the other hand, cavitation that causes enhanced perfusion has the disadvantage of 460
    potentially not eliciting a strong immune response due to lack of DAMP and TAA release. 461
    In these situations it is useful to know which bubble behaviour causes which vascular effect 462
    to ensure that the desired effect is achieved. 463

  3. It would be helpful to discuss potential future directions for research in this area. Suggesting novel experimental approaches or highlighting unanswered questions would encourage further investigation and advancement of the field.

    The lines 430-439 (copy-pasted above) suggests what future studies could focus on such as looking more into cavitation effects on cytokines and cancer associated fibroblasts. It also discusses how including more perfusion data throughout the study, for instance immediately before tumour excision, can advance our understanding of  the immunological response (lines 453-463, copy pasted above). Lastly, the discussion also describes how passive cavitation detection can be used to uncover a potential relationships between the oscillation regime and immunological effect (lines 464-474).

    Notably, bubble behaviour is not only dependent on US parameters such as frequency, 464
    PNP, pulse repetition frequency, and sonication duration, but also the characteristics of the 465
    CAs, such as their size, composition, and concentration[74]. Additionally, the tumour type, 466
    location, and environment will affect the bubble oscillation and the subsequent physical 467
    mechanisms. Thus, passive cavitation detection, i.e. monitoring cavitation behaviour, is an 468
    essential tool in terms of uncovering which bubble behaviour is dominant and desirable 469
    for certain biological and immunological effects[32,75–77]. Yet, only one paper in section 2 470
    assessed cavitation levels, allowing Bulner et al. to report which acoustic emissions were 471
    dominant[34], although Joiner et al. did confirm the presence of CAs in tumours after 472
    treatment by contrast enhanced imaging. Thus, it would be beneficial if future studies 473
    included passive cavitation detection to give more details about the oscillation regime. 474

  4. If the author inserts schematic diagrams of typical examples in the article, it will greatly help readers understand.

    A figure summarising the different immunological effects of cavitation have been added (Figure 1, see attachment). 

Reviewer 3 Report

This is a well-organized review article covering basic knowledge and current protocols/findings on cavitation-mediated immunomodulation and its use with ICIs. This reviewer strongly recommends the authors to prepare and add a schematic figure that summaries acoustic cavitation-mediated effects on TME and ICIs, including DAMPs, TAA/TSAs, and various types of innate and adaptive immune cells. Such figure would help a wider population of readers understand the essence of this review article.

Author Response

A figure summarising the immunological effects of cavitation have been added (Figure 1, see attachment).

Round 2

Reviewer 1 Report

The revised manuscript has made a great improvement. I have no more comments and recommends.

 Minor editing of English language required

Reviewer 2 Report

The authors well answered the raised questions. The manuscript can be published with this version.

Minor editing of English language required